# Mechanisms of Resistance to Immunotoxins Containing *Pseudomonas* Exotoxin A in Cancer Therapy

**DOI:** 10.3390/biom10070979

**Published:** 2020-06-30

**Authors:** Michael Dieffenbach, Ira Pastan

**Affiliations:** Laboratory of Molecular Biology, Center for Cancer Research, National Cancer Institute, National Institutes of Health, Bethesda, MD 20892-4264, USA; michael.dieffenbach@nih.gov

**Keywords:** immunotoxin, *Pseudomonas* exotoxin A, immunotoxin resistance, combination therapy

## Abstract

Immunotoxins are a class of targeted cancer therapeutics in which a toxin such as *Pseudomonas* exotoxin A (PE) is linked to an antibody or cytokine to direct the toxin to a target on cancer cells. While a variety of PE-based immunotoxins have been developed and a few have demonstrated promising clinical and preclinical results, cancer cells frequently have or develop resistance to these immunotoxins. This review presents our current understanding of the mechanism of action of PE-based immunotoxins and discusses cellular mechanisms of resistance that interfere with various steps of the pathway. These steps include binding of the immunotoxin to the target antigen, internalization, intracellular processing and trafficking to reach the cytosol, inhibition of protein synthesis through ADP-ribosylation of elongation factor 2 (EF2), and induction of apoptosis. Combination therapies that increase immunotoxin action and overcome specific mechanisms of resistance are also reviewed.

## 1. Introduction

Immunotoxins are a promising class of cancer therapeutics that combine the potent cytotoxicity of a toxin such as *Pseudomonas* exotoxin A (PE) with the selectivity of an antibody fragment or cytokine [1]. PE-based immunotoxins present several advantages over chemotherapy in that they kill cells by a unique mechanism (inhibition of protein synthesis) and they do not cause mutations by damaging DNA. Inhibition of protein synthesis kills cells by impairing survival mechanisms that require new protein expression. Target cancer antigens that have been investigated include CD22 [2], mesothelin [3], CD25 [4], CD30 [5], CD33 [6], Glypican 3 [7], carcinoembryonic antigen [8], and receptors for EGF [9], IL4 [10], IL6 [11], and IL13 [12]. Immunotoxins have been successful in treating certain cancers, such as drug resistant hairy cell leukemia [13]. However, the success of immunotoxins in the treatment of other cancers has been limited in part by the presence or emergence of immunotoxin resistance.

This review discusses cellular mechanisms of resistance to PE-based immunotoxins as well as current efforts to overcome them. While factors such as immunogenicity and the challenges of delivery to solid tumors also significantly hinder immunotoxin therapy, they are beyond the scope of this review. It should also be noted that resistance mechanisms to immunotoxins utilizing toxin domains from sources other than PE would typically vary depending on the cellular itinerary and toxin mechanism involved. However, when immunotoxin mechanisms overlap (particularly in the binding and internalization steps), resistance mechanisms may overlap as well.

## 2. Mechanism of PE-based Immunotoxins

The pathway that PE-based immunotoxins employ to enter and kill cells is complex and incompletely understood. As many resistance mechanisms interfere with specific steps, a brief description of the pathway is provided here. Using an immunotoxin targeting mesothelin as a model, the first step is binding of an immunotoxin to a receptor on the cell surface (Figure 1), followed by internalization into the endosomal compartment (Step 1).

The immunotoxin must then undergo processing and trafficking to reach the cytosol (Step 2). In the endosome, the immunotoxin is processed by the protease furin to separate the antibody fragment from the toxin. The antibody fragment goes to the lysosome where it is degraded. In contrast, the toxin is transferred to the Golgi region where the REDL peptide sequence at its C terminus binds to the KDEL receptor and brings it into the endoplasmic reticulum. How the toxin gets from the endosome into the Golgi and avoids transfer to lysosomes is not known, but the KDEL receptor could have a role in this process. The toxin then enters the cytosol. A prevailing theory is that the toxin unfolds in the ER to mimic an unfolded secretory protein to be exported by the Sec61p translocation pore [14].

After reaching the cytosol, the toxin binds to elongation factor 2 (EF2) and catalyzes the NAD dependent ADP-ribosylation of the diphthamide residue of EF2 (Step 3). Diphthamide is a modified histidine, which is found only on EF2. The presence of diphthamide is essential for the toxin to arrest protein synthesis because ADP-ribosylation of the diphthamide residue inhibits EF2 dependent chain elongation and synthesis of new proteins [15].

Inhibition of protein synthesis then triggers apoptotic signaling (Step 4). Apoptosis is a complex process involving a delicate balance between pro-apoptotic and anti-apoptotic proteins. How protein synthesis inhibition results in apoptosis is not completely clear, but it relies in part on the presence of the pro-apoptotic protein Bim, the depletion of the rapidly degraded pro-survival protein Mcl-1, and the subsequent activation of the pro-apoptotic protein Bak [16,17]. Bak then induces the intrinsic pathway of apoptosis as evidenced by the activation of caspase-3, -8, and -9, poly (ADP-ribose) polymerase (PARP) cleavage, and a decrease in mitochondrial membrane potential. Multiple apoptotic pathways may be involved—cathepsin B/L also plays a role in immunotoxin-mediated cell death through a caspase-independent process [18].

The pathway described above—(1) antigen binding and internalization, (2) processing and trafficking, (3) ADP-ribosylation of EF2 and inhibition of protein synthesis, and (4) induction of apoptosis—represents our current understanding of the direct effects of immunotoxin action. It should be noted that there are gaps in existing knowledge, for example, many of the proteins necessary for immunotoxin trafficking are still poorly characterized. Additionally, there is evidence of other indirect but significant effects of immunotoxin action, which may also be involved in mechanisms of resistance. One paper indicated that an early effect of PE immunotoxins might be the partial depletion of cellular ATP, which induces a cytoprotective response involving AMPK, JNK, and p38MAPK [19]. As with the pro-survival proteins such as Mcl-1, this response must be overwhelmed for apoptosis to occur. Protein synthesis inhibition can also have varied and far-reaching effects. Interestingly, a 2018 study by El-Behaedi et al. found that 48-hr treatment of pancreatic cancer cells with LMB-100 (a mesothelin-targeting immunotoxin) did not significantly alter total protein levels, despite complete inhibition of protein synthesis. Instead, it preferentially decreased levels of rapidly degraded proteins that included many oncogenic signaling molecules and growth factors (such as a vascular endothelial growth factor) secreted by cancer cells [20]. This suggests that beyond directly killing target cells, immunotoxins may impact the tumor microenvironment by reducing the release of these tumor-stimulating factors.

Resistance can potentially occur at any step in the immunotoxin pathway, and an improved understanding of the immunotoxin function may reveal new resistance mechanisms. Known mechanisms of resistance at each step will be discussed in greater detail below, along with existing strategies to combat resistance.

## 3. Antigen Binding and Internalization (Step 1)

Immunotoxin molecules that successfully reach cancer cells in a human tumor must first bind to the target antigen and then be internalized. Thus, resistance mechanisms can include decreased antigen expression. Mesothelioma cells taken from patients often rapidly lost mesothelin expression when cultured, even for as few as three passages. This loss of expression was correlated with resistance to SS1P, a mesothelin-targeting immunotoxin [21]. Downregulation of target antigens such as mesothelin can occur through methylation of the mesothelin promoter [22]. Signaling pathways can also influence the antigen expression. The tyrosine kinase DDR1 has been found to regulate surface expression of mesothelin through interactions with collagen. DDR1 silencing in A431/H9 cells resulted in increased expression of mesothelin and increased internalization of the mesothelin-targeting immunotoxin RG7787 [23]. Additionally, combination therapies may increase or decrease levels of the target antigen; for example, exposure to sublethal radiation appears to increase mesothelin cell-surface expression [24] and results in increased sensitivity to SS1P in nude mice with mesothelin-expressing tumors [25]. Resistance through loss of antigen impacts treatment considerations—the ideal antigen should be essential for cancer cell proliferation and therefore not easily lost, and measurement of the amount of antigen expression in patient samples could help predict the effectiveness of immunotoxin treatment.

Some antigens (as in the case of mesothelin) may be shed from the cell surface and released into the bloodstream or extracellular environment, where they can sequester immunotoxins and prevent them from reaching target cells. Factors that increase antigen shedding can result in resistance. For example, the transmembrane glycoprotein TACE has been implicated in mesothelin shedding and resistance to the mesothelin-targeting immunotoxin SS1P. Knockdown with siRNA or small molecule inhibition of TACE increases sensitivity of cultured cells to SS1P [26], and similar results were seen in an engineered cell line with a mutant form of mesothelin lacking the TACE cleavage site [27]. The tubulin-targeting chemotherapeutic paclitaxel (Taxol) appears to synergize with SS1P partly by reducing levels of shed mesothelin in paclitaxel-sensitive (but not paclitaxel-resistant) KB tumors in mice [28]. This led to a phase I trial of nab-paclitaxel (albumin-bound paclitaxel) in combination with the immunotoxin LMB-100 in advanced pancreatic adenocarcinoma [29].

## 4. Immunotoxin Processing and Trafficking (Step 2)

Immunotoxins follow a complex route after internalization, and many factors during processing and trafficking can contribute to resistance. Cleavage of the toxin fragment from the antibody fragment by the protease furin is a vital step, and the Daudi cell line (a Burkitt’s lymphoma line) was shown to be resistant to an anti-CD22 PE-based immunotoxin, possibly due to impaired cleavage by furin. Prior partial cleavage of the immunotoxin with furin was shown to overcome this resistance in Daudi cells [30]. Interestingly, immunotoxins targeting other receptors on the same cell line did not demonstrate the same resistance [31]. This suggests that different intracellular trafficking routes may increase or decrease levels of exposure to furin, or that steric effects due to the structure of the targeting fragment may aid or impede access to the furin cut site. The furin-deficient LoVo colon cell line is resistant to TGFα-PE38, an immunotoxin composed of transforming growth factor (TGF) linked to a PE fragment. Once again, pretreatment of the immunotoxin with furin in vitro increased toxicity in LoVo cells [31].

It has also been observed that knockdown of the insulin receptor increases cleavage of SS1P by furin through an unknown mechanism and correspondingly increases SS1P cytotoxicity by increasing the amount of processed toxin that can reach the cytosol and inactivate EF2 [32]. The insulin receptor is a tyrosine kinase, which is activated by insulin and insulin-like growth factor 1 to stimulate cell proliferation and survival pathways. Surprisingly, however, insulin receptor knockdown appeared to only affect immunotoxin processing and not levels of pro- or anti-apoptotic proteins as might be expected due to the role of the insulin receptor in cancer cell survival. The same study tested several compounds known to inhibit insulin activity, but none replicated the increase in SS1P activity observed with insulin receptor knockdown. This suggests that the insulin receptor may regulate immunotoxin action through a mechanism distinct from its insulin-associated kinase activity, although the exact mechanism remains to be identified.

To determine if other tyrosine kinases might also regulate immunotoxin activity, a separate study was performed to follow up on this result. The 87 known tyrosine kinases in addition to the insulin receptor were screened with siRNA. Several siRNA knockdowns (*HCK, SRC, PDGFRβ,* and *BMX*) were found to sensitize cells to SS1P. Knockdown of the Src family member *HCK* produced the largest effect and was found to increase furin cleavage similarly to the insulin receptor knock down. In addition to this, *HCK* knockdown also affected later stages of the immunotoxin pathway by decreasing Mcl-1 and raising Bax. Src inhibitors, such as Bosutinib, were able to replicate the effect of *HCK* knockdown in both cell culture and mouse xenograft models [33]. As Bosutinib is already approved to treat myelocytic leukemia, it was suggested that combination treatment with Bosutinib and HA22 (moxetumomab pasudotox) might be effective in some HA22-resistant acute lymphoblastic leukemia (ALL) patients.

A variable portion of immunotoxin molecules that enter cells are routed to the lysosome and destroyed [34]. This appears to depend in part on the itinerary taken through the cell. In a 2012 study, Tortorella et al. used Chinese hamster ovary cells expressing CD25 fused to either TGN38 or furin, two transmembrane proteins that are brought to the trans-Golgi network through different established routes [35]. When these cells were exposed to the anti-CD25 immunotoxin LMB-2, the intracellular route taken by LMB-2 bound to chimeric TGN38 resulted in more efficient delivery of the toxin to the cytosol and ultimately in greater cytotoxicity. This suggests that some resistance due to immunotoxin destruction in lysosomes may be overcome through antigen choice and immunotoxin design. Agents that disrupt lysosomal function or integrity may also increase immunotoxin action. An early study indicated that the lysosomotropic agent β-glycylphenylnaphthylamide enhanced immunotoxin action in this way. The same study as well as a later follow-up study also found that the calcium-channel blocker verapamil and several of its analogues could enhance immunotoxin action. Although the mechanism was unclear and did not seem to relate to blocking of calcium channels, it was shown that these drugs delayed lysosomal degradation of immunotoxins [36,37].

To determine if other genes regulated immunotoxin resistance, a whole-genome siRNA knockdown screen was performed by Pasetto et al. that revealed several ER/Golgi proteins were involved. Knock-down of these genes, which included *COPB1*, *COPE*, *SLC33A1*, and two ARF-related proteins (*ARF4* and *ARL1*), sensitized cells to SS1P [38]. This implies that there are natural inefficiencies in the immunotoxin trafficking pathway, and that the normal function of these proteins reduces immunotoxin efficacy. While the precise role of these proteins in mitigating toxicity remains to be investigated, it is possible that overexpression of these proteins or modifications to other ER/Golgi proteins could contribute to resistance.

Another significant step in the intracellular trafficking of PE-based immunotoxins involves binding to the KDEL receptor to mediate transport from the Golgi to the ER [39]. Targeting the KDEL receptor with microinjected antibodies causes resistance to PE by blocking transport to the ER and preventing the toxin from reaching the cytosol [40]. Knockdown of KDEL receptor 2 also resulted in resistance in the whole-genome siRNA screen described above [39]. However, whether KDEL receptor loss occurs in a clinical setting is unknown.

Agents that enhance immunotoxin transport/release to the cytosol can also sensitize cells to immunotoxin killing. One promising such agent is ABT-737, a BH3 mimetic. In addition to its pro-apoptotic role (discussed below), ABT-737 was found by Traini et al. in 2010 to increase ER stress and facilitate release of immunotoxins from the ER to the cytosol. When tested in combination with various inhibitors of protein synthesis (diphtheria toxin, cycloheximide, HB21-PE40 (a PE immunotoxin targeting the transferrin receptor), and HB21-CET40 (an immunotoxin derived from cholera toxin)), synergistic killing greater than 10-fold was only observed with agents that pass through the ER to the cytosol (the PE and CET immunotoxins). Combination of ABT-737 with two different PE immunotoxins produced a 25-fold reduction in protein synthesis compared to immunotoxin alone, suggesting that ABT-737 may increase the number of immunotoxin molecules reaching the cytosol [41]. Another study by Risberg et al. in 2011, tested ABT-737 in combination with an immunotoxin targeting the high molecular weight-melanoma associated antigen in melanoma cell lines and demonstrated comparable synergy, though the combination was only able to produce growth delay in a mouse model [42]. Further analysis revealed that treatment with ABT-737 resulted in significantly increased expression of the ER stress marker ATF4 in addition to increased permeability of the ER membrane to soluble ER proteins as well as PE-based immunotoxins [43]. The combination was also effective in a prostate cancer model with an immunotoxin targeting the prostate-specific membrane antigen [44]. Most recently, ABT-737 and two other BH3 mimetics were tested in combination with a chondroitin sulfate proteoglycan 4 (CSPG4)-targeting immunotoxin in patient-derived melanoma, glioblastoma, and breast cancer cell lines and demonstrated significant synergy. The same study further confirmed the efficacy of the combination in in vivo nude mice studies of glioblastoma and metastatic cerebral melanoma [45].

Resistance to many conventional anticancer drugs occurs through the action of transporters such as the multidrug resistance-associated protein (MRP) family, which pump a variety of exogenous molecules out of the cell [46]. One study has shown that overexpression of MRP1 (which is frequently observed in multi-drug resistant cancer cells) causes resistance to an IL-4–PE fragment immunotoxin, though overexpression of MRP2-5 or the P-glycoprotein transporter had no effect. Overexpression of MRP1 did not cause resistance to native PE in the same cells, implying that the IL-4 portion of the immunotoxin may have been the target of MRP1 and that the resistance is not generalized to all PE-immunotoxins. However, this demonstrates that the choice of the antibody/cytokine fragment of an immunotoxin may result in distinct mechanisms of resistance [47].

## 5. Inhibition of Protein Synthesis (Step 3)

Upon arrival in the cytosol, PE-based immunotoxins catalyze the ADP-ribosylation of the diphthamide residue of EF2, resulting in inhibition of protein synthesis and ultimately apoptosis. Several resistance mechanisms have been identified in this process.

Diphthamide is a unique post-translationally modified histidine at position 715 in human EF2, believed to play a role in translational fidelity (especially for proteins containing selenocysteine) [48]. Biosynthesis of diphthamide involves seven genes, *DPH1-7* [49]. Although highly conserved, diphthamide is not crucial for EF2 function and several PE-resistant cell lines have been isolated with diphthamide biosynthetic mutations. These include methylation of the *DPH1* and *DPH4* promoters, as well as deletion of *WDR85*/*DPH7* [50,51,52]. Resistance through methylation can be partially overcome with the DNA methylation inhibitor 5-azacytidine [50,51,52]. Another study used ZFN mutagenesis to produce heterozygous or homozygous knockouts of DPH genes in MCF7 cells. Heterozygous knockouts exhibited no altered phenotype, while homozygous knockouts of *DPH3*, *6*, and *7* were lethal. Knockouts of *DPH1*, *2*, *4*, and *5* were viable and resistant to PE. Interestingly, loss of diphthamide seemed to partially activate the NF-κB and death receptor pathways, although not enough to activate apoptosis alone [53].

The relevance of impaired diphthamide synthesis in immunotoxin resistance in patients was studied by Müller et al. When Moxetumomab pasudotox-resistant primary ALL lines from pediatric patients were examined, cells from two of six patients were found to have reduced expression of *DPH4*. However, reduced diphthamide synthesis gene expression was also observed in immunotoxin-responsive patient cells. It was concluded that diphthamide gene expression could not account for the majority of resistant cases [54]. Mouse models injected with GFP/luciferase-expressing KOPN-8 or Reh cells revealed that a few resistant cells persisted in the bone marrow and subsequently expanded systemically. Resistant cells could not be cultured in the absence of stromal cells and showed no decrease in diphthamide gene expression, though the expression of CD22 as well as other B cell markers was reduced. Significant chromosomal alterations were also observed. Interestingly, 5-azacytidine prevented this resistance in one mouse model and delayed it in another, though it was unable to overcome established resistance [54]. 

The collagen-activated tyrosine kinase DDR1 has also been shown to play a role in PE-based immunotoxin resistance at the level of protein synthesis. Knockdown or inhibition of DDR1 sensitized several cell lines to the immunotoxin RG7787, and subsequent investigation revealed that DDR1 silencing decreased the expression of several ribosomal proteins (including RPL10A, RPL22, RPL24, and RPL38) in KB and A431/H9 cells and inhibited protein synthesis. Direct silencing of RPL10A or RPL38 also inhibited protein synthesis, an effect that was increased by RG7787 administration. In the same study, overexpression of DDR1 was able to confer up to 5-fold resistance to RG7787. Culturing cells in the presence of collagen also resulted in resistance, presumably through activation of DDR1. Combination treatment with RG7787 and 7rh, a selective DDR1 inhibitor, resulted in enhanced cell killing [23].

## 6. Induction of Apoptosis (Step 4)

Protein synthesis inhibition alone is not sufficient to kill cells, as cells with impaired apoptotic pathways can survive even with complete protein synthesis inhibition [16]. PE immunotoxin-induced protein inhibition results in the loss of rapidly degraded proteins such as the anti-apoptotic protein Mcl-1, which binds and inhibits Bak (a mediator of the intrinsic apoptotic pathway) [16]. In addition to Mcl-1, the pro-apoptotic Bcl-2 proteins Bim and Noxa are also rapidly degraded and depleted upon protein synthesis inhibition, and Bim KO cells are completely resistant to PE immunotoxin treatment in vitro and in vivo [17]. In several epithelial, leukemia, and lymphoma lines, Bim expression levels below a certain threshold resulted in complete depletion of Bim and subsequent resistance upon immunotoxin treatment [17]. In patient samples as well, Bim expression correlated with a better response to immunotoxin treatment [17]. Overexpression of Mcl-1 and silencing of Bak have both been shown to result in resistance to PE in mouse embryo fibroblasts, despite comparable levels of protein inhibition to PE-sensitive cell lines [16]. A later study demonstrated similar results with SS1P and human pancreatic cancer cell lines with naturally low Bak levels (such as HTB-80 and Panc 3.014) [55]. In summary, either high levels of anti-apoptotic proteins or low levels of pro-apoptotic proteins can result in immunotoxin resistance, and such mechanisms are often observed in cancer cells.

While the exact mechanism has not been determined, the cellular apoptosis susceptibility gene (*CAS/CSEL1*) may also play a role in immunotoxin-induced apoptosis. *CAS* was identified by a 1995 study in which a cDNA expression library was used to isolate plasmids that provided resistance to PE-based immunotoxins in MCF-7 breast cancer cells [56]. One such plasmid encoded an antisense cDNA fragment of the *CAS* gene, which is homologous to the chromosome segregation gene *CSE1* in yeast. Further investigation revealed that reduction of CAS protein levels through expression of the antisense cDNA also resulted in resistance to native PE, diphtheria toxin, and tumor necrosis factor (TNF) α and β, though it did not cause resistance to other agents including the protein synthesis inhibitor cycloheximide. As CAS reduction did not measurably affect ADP-ribosylation of EF2 by PE/diphtheria toxin/PE-based immunotoxins or binding of TNF, it was suggested that CAS reduction mediates resistance by inhibiting specific pathways of apoptosis [57]. Evidence from other studies indicates that CAS expression is induced by various apoptotic stimuli and that CAS participates in the intrinsic apoptotic pathway by enhancing nucleotide exchange on the Apaf-1/cytochrome c complex, allowing the complex to oligomerize into the caspase-9-activating apoptosome [58]. This mechanism has been confirmed in TNF resistance [59], though not yet in immunotoxin resistance.

Resistance to apoptosis can be overcome by targeting multiple apoptotic pathways or by inhibiting overexpressed anti-apoptotic proteins. Du et al. 2011 found that using TRAIL or HGS-ETR2 (an anti-TRAIL receptor 2 agonist antibody) could overcome resistance associated with low Bak levels by activating the extrinsic pathway of apoptosis, which does not depend on Bak. Treatment with both SS1P and TRAIL/HGS-ETR2 caused synergistic, mitochondria-dependent cell death through caspase-8 activation [55]. Panbinostat, a pan-histone deacetylase inhibitor used in the treatment of multiple myeloma, also activates the extrinsic pathway and shows a similar synergistic effect with LMB-100 [60]. The BH3 mimetic ABT-737 also shows enhanced cell killing when combined with SS1P. ABT-737 binds and inhibits several BCL2 family anti-apoptotic proteins with the exception of Mcl-1, which is depleted through SS1P [61]. A related BH3 mimetic, ABT-263, showed similar synergy with the transferrin receptor-targeting PE immunotoxin HB21-PE40 in four small cell lung cancer lines [62].

## 7. Other Synergistic Agents

Some synergistic agents act through unclear mechanisms or separately affect multiple steps in the PE immunotoxin pathway. These agents reflect the current holes in our understanding of the immunotoxin mechanism and often in the mechanism of the synergistic agent. For example, the kinase inhibitor H89 was found to both increase PE immunotoxin-induced ADP-ribosylation of EF2 (through an unidentified mechanism occurring between immunotoxin internalization and enzymatic action) and to decrease levels of Mcl-1 in ALL cell lines (including patient-derived cells) and KB31 cells [63]. The decrease in Mcl-1 does not appear to be merely a consequence of increased delivery of the toxin to the cytosol, as treating cells with H89 alone moderately decreased Mcl-1 but did not measurably alter EF2. While H89 is considered to be primarily an inhibitor of protein kinase A with modest inhibition of several other kinases, other PKA activators and inhibitors showed no effect. S6K1, another kinase known to be inhibited by H89, appeared to be partly implicated. H89 inhibited S6K1 and prevented the phosphorylation of one of its targets, GSK3β [64]. Phosphorylation of GSK3β by S6K1 inhibits phosphorylation of Mcl-1 by GSK3β, which then induces ubiquitination and degradation of Mcl-1 [63]. A specific inhibitor for S6K1 replicated the decrease in Mcl-1 levels but only weakly enhanced ADP-ribosylation of EF2 and overall immunotoxin activity, suggesting that other unidentified targets of H89 may also contribute [63].

Actinomycin D demonstrates significant synergy with the PE-derived immunotoxin RG7787. While the mechanism of this synergy appears to be complex and to involve multiple genes/pathways, the primary effect appears to occur at the level of apoptosis. Actinomycin D in combination with RG7787 significantly increased levels of cleaved caspase-3, -8, and -9 as well as PARP, possibly through the NF-κB pathway. It also increased levels of p53 in the nucleus of KLM-1 cells, contributing to stress-induced cell death [65].

A study by Andersson et al. in 2006 identified the AMPK and JNK signaling pathways as initial stress responses activated by PE-based immunotoxins. These pathways were not activated by other protein synthesis inhibitors such as cycloheximide, although the exact mechanism by which immunotoxins activate these responses is not clear. Andersson et al. found that activation of these pathways delays but typically does not prevent immunotoxin-induced cell death, while treatment with the AMPK inhibitor compound C and the JNK inhibitor SP600125 synergistically sensitized cells to immunotoxin treatment. These represent additional candidates for combination therapy [19].

To identify other synergistic agents, Antignani et al. performed a chemical screen using the MIPE-3 compound library in combination with the immunotoxins SS1P and HA22 in KB3-1 and Nalm-6 cells. The screen identified multiple enhancers of immunotoxin activity, most notably the tyrosine kinase inhibitor nilotinib, the Wnt/Beta-catenin antagonist salinomycin, and the mTOR inhibitor everolimus. Compounds that antagonized immunotoxin activity were also identified and included PARP inhibitors, which appeared to directly inhibit the enzymatic activity of PE and PE-based immunotoxins [66]. This demonstrates that combination therapies must be chosen carefully to avoid antagonistic effects.

## 8. Conclusions

Mechanisms of resistance have been linked to nearly every known step of the PE-based immunotoxin pathway (a summary is provided in Table 1). Identification of those mechanisms has, in turn, provided clues to the rational development of new strategies to overcome resistance. Ideally, characterization of resistance phenotypes in a patient sample could be used to tailor immunotoxin therapy and improve the response. However, there are still many gaps in our knowledge. Portions of the pathway used by PE-based immunotoxins (such as the trafficking pathway and the induction of apoptosis following protein synthesis inhibition) are still not perfectly understood. Many of the resistance mechanisms described above have been characterized in cell lines or animal models, but the frequency of their occurrence in a clinical setting is less clear. As our understanding of immunotoxin resistance develops further, improved immunotoxin design and combination therapy will result in greater clinical success.

## Figures and Tables

**Figure 1 biomolecules-10-00979-f001:**
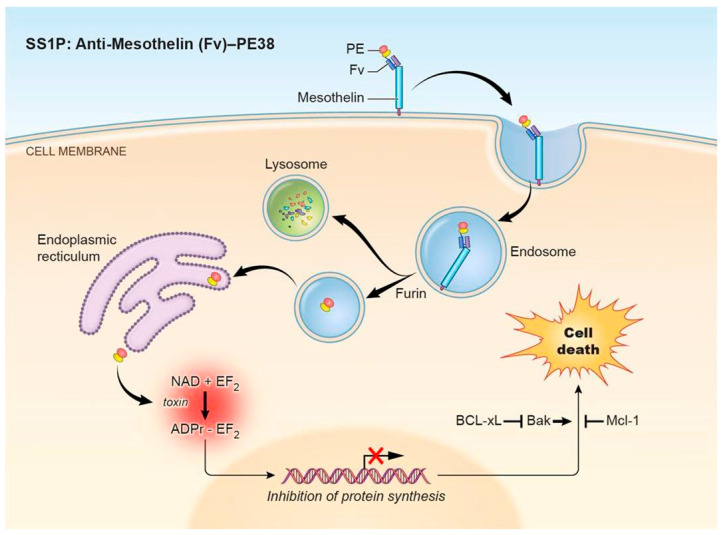
*Pseudomonas* exotoxin A (PE)-immunotoxin pathway. PE-based immunotoxins such as the mesothelin-targeting immunotoxin SS1P or LMB-100 act through a pathway involving (1) binding to the target antigen and internalization, (2) processing and trafficking, (3) inhibition of protein synthesis, and (4) induction of apoptosis. Abbreviations: PE: *Pseudomonas* exotoxin A; Fv: fragment variable; PE38: truncated PE fragment; EF2: elongation factor 2.

**Table 1 biomolecules-10-00979-t001:** Summary of factors resulting in immunotoxin resistance and combination therapies to overcome them.

Pathway Step:	Resistance Mechanism:	Combination Therapies and Strategies to Overcome Resistance:	References:
Binding to antigen and internalization	Antigen shedding	Paclitaxel (reduces shed mesothelin)TACE inhibitors (reduces shed mesothelin)	[26,27,28,29]
	Decreased antigen expression	Target multiple antigens, modulate antigen expression if possible	[21,22,23,24,25]
Trafficking and processing	Impaired cleavage by furin	Bosutinib (Src inhibitor)	[30,31,32,33]
	Lysosomal destruction	Optimize antigen choice, modulate lysosomal activity	[35,36,37]
	Trafficking inefficiencies	ABT-737	[38,40,41,42,43,44,45]
Protein synthesis inhibition	Loss of diphthamide residue	5-azacytidine (to reverse methylation of diphthamide synthesis gene promoters)	[50,51,52,53,54]
Induction of apoptosis	Low levels of Bak, Bim, or CAS	Activators of the extrinsic apoptotic pathway (panbinostat, TRAIL, etc.)	[16,17,55,56,57]
	High levels of Mcl-1		[16]

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
