# Peer review of "Mechanisms of Resistance to Immunotoxins Containing Pseudomonas Exotoxin A in Cancer Therapy"

_biomolecules, 2020, doi:10.3390/biom10070979_

Round 1

Reviewer 1 Report

The manuscript entitled “Mechanisms of resistance to immunotoxins containing Pseudomonas exotoxin A in cancer therapy” is a very nice review focused on the most studied immunotoxins, those made with PE. Moreover, authors carefully detail the existing literature related to the resistance of target cells to the effect of these immunotoxins. The structure of the manuscript makes very clear to follow the different resistance mechanisms depending on the process involved during immunotoxin binding, internalization and cytosolic activity. Finally, as far as this reviewer is concerned, no other review so well focused on this aspect has been published so far, and considering the relevance of recognizing and overcoming these resistance mechanisms, the manuscript is well suited for publication in Biomolecules Journal.

This reviewer has just a couple of comments that might be mentioned in the manuscript:

In relation to the combined therapies to overcome resistance, nothing is mentioned about unwanted effects due to synergistic mechanisms. Are there examples in this direction with these immunotoxins?

The manuscript focuses on PE-based immunotoxins but, are there comparative studies with other toxin made immunotoxins in terms of resistance?

Other minor comments:

  1. In line 27 authors describe immunotoxins as made of bacterial toxins. However, other trials with toxins of different nature like fungal ribotoxins are being developed.
  2. Also in line 32 authors might want to include the CEA antigen as an investigated target not only with PE based immunotoxins but also with other different toxic modules.

Author Response

  • Based on Reviewer 1’s suggestion to discuss any unwanted effects from combination therapies, we referenced a screen that identified compounds that demonstrate unwanted or antagonistic effects with immunotoxins (lines 339-345, “To identify other synergistic agents…”).
  • A brief comparison of resistance to non-PE-based immunotoxins was addressed per Reviewer 1’s comments (lines 44-47, “It should also be noted…”).
  • In response to Reviewer 1’s point that not all immunotoxins are derived from bacterial toxins, removed the word “bacterial” in defining immunotoxins and describing the origin of the toxin moiety (line 31).
  • As requested, added the CEA antigen to the list of example antigens targeted by immunotoxins (lines 36-37).

Reviewer 2 Report

In recent years, PE-based immunotoxins have been developed for clinical application. In the course of this development, various resistance mechanisms have become apparent which may reduce their efficacy. These mechanisms are clearly summarized and discussed in this article. Furthermore, different strategies are shown how these resistance mechanisms can be overcome.
The review provides a good overview, is comprehensive and easy to read. In some sections it is necessary to include more references. Overall, I recommend an acceptance after minor revision.

Points of criticism are:

1.) Lines 129-130: The authors discuss the furin cleavage as critical step for resistance and state: „Interestingly, immunotoxins targeting other receptors on the same cell line did not demonstrate the same resistance, which suggests that different intracellular trafficking routes may increase or decrease levels of exposure to furin [19]“. Insufficient furin cleavage may, however, also be due to inaccessibility of the furin cleavage site in the immunotoxin molecule to the enzyme furin, depending on the structure of the antigen-binding domain. This should also be discused in the text.

2.) References should be added to the following sentences:

Lines 26-27: Immunotoxins are ….

Lines 31-32: Target cancer antigens …

Lines 155-156: A variable portion of …

Lines 177-178: Another significant step …

Lines 204-106: Resistance to many …

Line 219: Biosynthesis of diphtamide…

Lines 229-233: The frequency of clinical resistance… - It was conducted that …

Line 252: Protein biosynthesis inhibition …

Line 252: PE immunotoxin-induced…

Line 254: In addition to Mcl-1…

Line 257: In several epithelial…

Line 295: For example, the kinase inhibitor …

3.) For a better overview, numbers for the 4 steps of the intoxication pathway should be included in Figure 1.

4.) An abbreviation list should be added.

5.) Abbreviations should be added to the legend of Figure 1.

6.) Line 83: …(„such as Vascular Endothelial Growth Factor or VEGF“)… is misleading and should be corrected.

7.) Line 119: …“in tumors in mice“… is repeated in the sentence and should be deleted.

Author Response

  • As Reviewer 2 observed, there is the possibility that the structure of the targeting domain may be responsible for differences in furin cleavage between different immunotoxins (lines 141-143). We have included this point in the text.
  • Additional citations were added at the points specified by Reviewer 2 (lines 36-37, 170, 221, 234, 237, 250, 269, 272, 274, 276, 315, and 345).
  • As requested, we added numbered pathway steps to figure 1 to provide a clearer overview.
  • An abbreviation list was added (lines 23-26).
  • As Reviewer 2 suggested, we also added an abbreviation list to the figure 1 legend (lines 102-103).
  • We removed “or VEGF” in line 91, as Reviewer 2 pointed out that the original wording could be interpreted to suggest that Vascular Endothelial Growth Factor and VEGF were separate examples.
  • We deleted the redundant usage of “in tumors in mice” (lines 129-130) noted by Reviewer 2.